# Exosomes as a Source of Biomarkers for Gastrointestinal Cancers

**DOI:** 10.3390/cancers15041263

**Published:** 2023-02-16

**Authors:** Jingjing Yu, Arsha Ostowari, Amber Gonda, Kiarash Mashayekhi, Farshid Dayyani, Christopher C. W. Hughes, Maheswari Senthil

**Affiliations:** 1Department of Surgery, University of California, Irvine Medical Center, Orange, CA 92868, USA; 2Division of Hematology/Oncology, Department of Medicine, University of California, Irvine Medical Center, Orange, CA 92868, USA; 3Department of Molecular Biology & Biochemistry and Department of Biomedical Engineering, University of California, Irvine, CA 92697, USA

**Keywords:** liquid biopsy, exosomes, biomarker, gastrointestinal cancers

## Abstract

**Simple Summary:**

Exosomes are small (40–160 nanometer) extracellular vesicles with significant roles in cancer development and progression. Exosomes are abundantly produced by cancer cells, carry tumor-specific content, such as DNA, RNA, and proteins, and have the potential to serve as biomarkers and therapeutic targets. Since exosomes are present in various biofluids, such as blood, saliva, urine, and peritoneal fluid, they render themselves as a great platform for the development of liquid biopsies. This review offers a comprehensive summary of diagnostic, prognostic, and predictive exosomal biomarkers in colorectal and gastric cancers. We also discuss the challenges and limitations of exosomes in clinical application and future prospects.

**Abstract:**

Exosomes are small, lipid-bilayer bound extracellular vesicles of 40–160 nanometers in size that carry important information for intercellular communication. Exosomes are produced more by tumor cells than normal cells and carry tumor-specific content, such as DNA, RNA, and proteins, which have been implicated in tumorigenesis, tumor progression, and treatment response. Due to the critical role of exosomes in cancer development and progression, they can be exploited to develop specific biomarkers and therapeutic targets. Since exosomes are present in various biofluids, such as blood, saliva, urine, and peritoneal fluid, they are ideally suited to be developed as liquid biopsy tools for early diagnosis, molecular profiling, disease surveillance, and treatment response monitoring. In the past decade, numerous studies have been published about the functional significance of exosomes in a wide variety of cancers, with a particular focus on exosome-derived RNAs and proteins as biomarkers. In this review, utilizing human studies on exosomes, we highlight their potential as diagnostic, prognostic, and predictive biomarkers in gastrointestinal cancers.

## 1. Introduction 

Over the last decade, there has been a paradigm shift in cancer management with more customized and dynamically adjusted treatment designs based on the tumor status in an individual patient. This advanced approach is made possible by liquid biopsies capable of early detection, identifying somatic gene alterations and monitoring the treatment response and tumor progression. Liquid biopsies have distinct advantages over conventional tissue biopsies due to their less invasive nature, lower costs, and the feasibility to be repeated multiple times during treatment and surveillance. Furthermore, they can be performed on blood and other biofluids such as urine, saliva, and ascites [1,2].

Circulating tumor DNA (ctDNA) is the main liquid biopsy currently in clinical use for the management of gastrointestinal (GI) cancers. ctDNA is single- or double-stranded DNA released from necrotic or apoptotic tumor cells and carries molecular information that can be used to guide clinical decisions [1]. However, ctDNA has important limitations as its detection is influenced by several factors including disease burden, disease location, treatment, and tumor vascularization. Specifically in GI cancers, the detection of ctDNA is affected by the type of tumor and location of metastases. Recent studies by our group and others have demonstrated that among patients with stage IV GI cancers, those with peritoneal carcinomatosis (PC) either have undetectable or significantly lower ctDNA levels compared with other metastatic sites, highlighting the limitations of ctDNA [3,4,5]. Hence, further work to develop alternate liquid biopsy tools that are reliable and informative are necessary.

As the field of liquid biopsy continues to expand and refine, the value of exosomes as an important alternate platform is being increasingly recognized. Exosomes are extracellular vesicles (EVs) surrounded by a lipid bilayer membrane that range in size between 40 and 160 nanometers [6]. Exosome biogenesis begins with invagination of the cell membrane, then continues with a tightly regulated process with active sorting and packaging of exosomal content (Figure 1) [6,7]. Exosomes contain a variety of substances such as lipids, proteins, DNA, messenger RNA (mRNA), short single-stranded microRNAs (miRNAs, 18–25 nucleotides (nt)), long non-coding RNAs (lncRNAs, >200 nt), and novel circular RNAs (circRNA) (Figure 1) [6,8,9]. Besides the common cargo, some of the contents of exosomes are specific to their cell of origin and can be used to identify the source [10].

The function of exosomes was originally described by Pan and Johnstone as they tracked the loss of transferrin receptors via released vesicles during reticulocyte maturation [11]. Previously regarded as “garbage bins”, a plethora of preclinical and human studies have now demonstrated the active role of exosomes in cancer intercellular communication [12]. Exosomes play key roles in cancer, including creation of the premetastatic niche, tumorigenesis, tumor progression, immune escape, treatment resistance, and signaling between tumor cells and the surrounding tumor microenvironment (Figure 2) [13,14]. Hence, the content of exosomes could potentially be used as biomarkers, particularly since exosomes are readily found in a variety of biofluids and are produced more by malignant cells than normal cells (Figure 3) [8,13].

Furthermore, when compared with ctDNA, exosomes offer several advantages. First, exosomes are released by all living cells and may reveal information about living tumor cells, unlike ctDNA, which is released through apoptosis or necrosis [15]. Because of their abundance, less blood volume is required for exosomes compared with ctDNA. Exosomes are also very stable under different storage conditions due to their lipid bilayer, which protects their DNA, RNA, and protein contents [1,16]. ctDNA, on the other hand, is rapidly cleared from blood and susceptible to degradation in circulation due to DNase activity [2]. Given the potential of exosomes as liquid biopsies, many studies have been conducted, particularly evaluating RNAs and proteins from exosomes. In this review, we summarize human studies focused on biofluids such as blood and peritoneal fluid that have investigated the utility of exosomes as diagnostic, prognostic, and predictive biomarkers in colorectal and gastric cancers (GCs). All of the studies described below compare patients with cancer to healthy controls (HCs) unless otherwise specified.

## 2. Colorectal Cancer

Colorectal cancer (CRC) is the third most diagnosed cancer in both men and women in the United States (US), with an estimated 106,180 new cases and an estimated 52,580 deaths in 2022 [17]. Early detection is key as 5-year survival for early stages is 90% but 13% for stage IV disease [18]. Exosomes are being studied as a less invasive source of diagnostic, prognostic, and predictive biomarkers for CRC, as summarized below. 

### 2.1. Diagnostic Biomarkers

The majority of the studies investigating exosome biomarkers for the diagnosis of CRC have predominantly focused on miRNAs. Table 1 provides a summary of the studies that have evaluated differentially expressed miRNAs, mRNAs, and lncRNAs between patients with CRC and HCs as potential diagnostic biomarkers. Ogata-Kawata et al. identified seven miRNAs (let-7a, miR-1229, miR-1246, miR-150, miR-21, miR-223, and miR-23a) in serum exosomes of CRC patients that were significantly overexpressed compared with HCs and downregulated following resection of the primary tumor, suggesting that the overexpressed miRNAs were of tumor origin [19]. Similarly, decreases in the levels of overexpressed RNAs following surgical resection of the tumor were observed by Ostenfeld et al. and Liang et al. [20,21]. Ostenfeld et al. in particular, attempted to evaluate tumor-derived EVs specifically by isolating epithelial cell adhesion molecular (EpCAM) positive EVs from the plasma of CRC patients and performing subsequent miRNA profiling. Thirteen miRNAs from these EpCAM^+^ EVs were overexpressed in CRC patients before surgical resection and eight of these miRNAs were downregulated after surgical resection, suggesting these miRNAs were of tumor origin [20]. Although the biomarkers were different between the three studies, the collective observations affirm that a portion of the overexpressed exosomal genes in the peripheral blood originate from the tumor and could provide meaningful insights about the tumor status.

Several miRNAs, including let-7a, miR-27a-3p, miR-383-5p, and miR-486-5p, that have well established roles in cancer have also been identified to be overexpressed in CRC patients [19,20,22,23]. Additionally, work has been performed to identify biomarkers that can be used to detect the early stages of CRC cancer. Wang et al. studied miR-125a-3p by comparing its expression in the plasma exosomes of patients with stage I and II colon cancer with HCs. mir-125a-3p expression was significantly increased in patients with early-stage disease. It is important to note that the predictive accuracy of miR-125a-3p to detect colon cancer improved from an area under the curve (AUC) of 0.685 to 0.855 when combined with the conventional diagnostic marker carcinoembryonic antigen (CEA) [24]. Similar observations of improved predictive accuracy when combining exosomal biomarkers with traditional tumor markers were observed in a study by Liu et al. In this study, exosomal lncRNA CRNDE-h was found to be higher in CRC patients compared with those with benign colon diseases and HCs (AUC = 0.892), and when combined with CEA, the diagnostic accuracy improved (AUC = 0.913) [25]. These observations open the possibility of utilizing exosome biomarkers as companion diagnostics with the existing modalities to improve diagnostic performance.

Other lncRNAs have also been identified as important diagnostic biomarkers of CRC. The lncRNA UCA1 has been shown to be downregulated in serum exosomes of patients with CRC, whereas the lncRNA TUG1 is upregulated compared with HCs. In combination, TUG1 and UCA1 had an AUC of 0.814 with a sensitivity of 93% and a specificity of 64% in distinguishing CRC patients from HCs [26]. Additionally, the lncRNA RPPH1 was found to have significantly higher expression in the plasma exosomes of CRC patients compared with HCs and the expression levels significantly decreased following surgical resection. The diagnostic power of RPPH1 to discriminate CRC patients from HCs (AUC = 0.856) was better than CEA (AUC = 0.790) [21].

Our group identified a gene signature that includes a combination of miRNA, mRNA and lncRNA. Plasma exosomes were isolated from patients with non-metastatic and metastatic (visceral metastases and PC) colon cancer and compared with HCs. Next-generation sequencing was used and after excluding highly prevalent and overlapping tRNA transcripts, 445 highly differentially expressed genes were identified. This gene signature, named ExoSig445, was able to fully discriminate colon cancer patients from HCs based on expression levels, suggesting gene panels can be developed as highly sensitive liquid biopsy tests (in press).

**Table 1 cancers-15-01263-t001:** Summary of diagnostic exosomal biomarkers for colorectal cancer.

Author(s)	Year	Biomarker(s)	Source	Findings
Ogata-Kawata et al. [19]	2014	7 miRNAs: let-7a, miR-1229, miR-1246, miR-150, miR-21, miR-223, miR-23a	Serum	7 miRNAs levels were higher in CRC patients and significantly decreased after removal of the primary tumor Highest predictive value of CRC: miR-23a (AUC = 0.953), miR-1246 (AUC = 0.948), miR-21 (AUC = 0.798)
Ostenfeld et al. [20]	2016	8 miRNAs: miR-16-5p, miR-23a-3p, miR-23b-3p, miR-27a-3p, miR-27b-3p, miR-30b-5p, miR-30c-5p, miR-222-3p	Plasma	All 8 miRNAs levels were higher in EpCAM^+^ exosomes of CRC patients and decreased following tumor resection, suggesting the miRNAs are of tumor origin
Dong et al. [27]	2016	mRNA KTTAP5-4, mRNA MEGEA3, lncRNA BCAR4	Serum	Exosomal expression of KRTAP5-4, MAGEA3 and BCAR4 was increased in CRC patients Combination of the three had AUC of 0.938 in predicting CRC
Wang et al. [24]	2017	miR-125a-3p	Plasma	Significantly higher levels in patients with stage I/II colon cancer miR-125a-3p with CEA improved prediction of colon cancer (AUC = 0.855) compared with miR-125a-3p (AUC = 0.685) or CEA (AUC = 0.836) alone
Liu et al. [23]	2018	miR-486-5p	Plasma	miR-486-5p exosomal expression level was higher in CRC patients across all stages (I-IV) (AUC = 0.713)
Barbagallo et al. [26]	2018	lncRNA UCA1, lncRNA TUG1	Serum	UCA1 was downregulated, whereas TUG1 was upregulated in CRC patients Combination of UCA1 with TUG1 AUC = 0.814
Karimi et al. [28]	2019	miR-301a, miR-23a	Serum	Higher exosomal miR-301a and miR-23a expression in CRC patients miR-301a AUC = 0.84; miR-23a AUC = 0.90
Liang et al. [21]	2019	lncRNA RPPH1	Plasma	Exosomal RPPH1 expression was significantly higher in CRC patients and significantly decreased following surgical resection Diagnostic power of RPPH1 (AUC = 0.856) was better than CEA (AUC = 0.790), CA 19-9 (AUC = 0.544) and CA 125 (AUC = 0.654)
Maminezhad et al. [29]	2020	6 miRNA signature: let-7a, miR-150, miR-143, miR-145, miR-19a, miR-20a	Serum	miR-19a, miR-20a, miR-150, and let-7a levels were significantly higher in CRC patients while miR-143 and miR-145 were significantly decreased AUCs ranged from 0.71 (let-7a) to 0.87 (miR-19a)
Vallejos et al. (in press)	2022	445 differently expressed genes from exosomal RNA (ExoSig445)	Plasma	ExoSig445 discriminated patients with non-metastatic and metastatic colon cancer from HCs 58 genes from ExoSig445 were overexpressed in colon cancer tissue compared with normal tissue using TCGA data

CRC = colorectal cancer, HC = healthy control, CEA = carcinoembryonic antigen, CA = cancer antigen, TCGA = The Cancer Genome Atlas.

### 2.2. Prognostic Biomarkers

Several of the biomarkers in CRC have both diagnostic and prognostic significance. The studies that have specifically reported the association of exosome biomarkers with prognosis have been summarized in Table 2. Some miRNA biomarkers have the potential to differentiate the early from late stages of CRC. In a study by Fu et al., overexpression of miR-17-5p and miR-92a-3p positively correlated with pathologic TNM stage, which allowed the authors to not only differentiate CRC from HCs but also non-metastatic disease from metastatic disease with high accuracy (AUC > 0.8) [30]. In a different study, plasma exosomal miR-21 was significantly higher in TNM stages III (*n* = 98) and IV (*n* = 67) compared with stages I (*n* = 51) and II (*n* = 110) [31]. Biomarkers that can aid in distinguishing early from late-stage CRC can be useful in guiding treatment, especially as neoadjuvant systemic therapy will often be considered for locally advanced and metastatic colon cancer [32].

Other studies have elucidated the role of miRNAs as prognostic biomarkers for recurrence. Liu et al. compared serum exosomes of patients with recurrent stage II/III disease with those without recurrent disease and found miR-4472-3p was upregulated in the recurrent disease group. Furthermore, miR-4772-3p was a predictor of recurrence with an OR of 11.3 (95% CI 2.38–53.2, *p* = 0.002) on multivariate logistic regression analysis and AUC of 0.72 based on the receiver operating characteristic (ROC) curve. Patients with higher miR-4772-3p expression had a shorter time to recurrence compared with lower miR-4772-3p expression (32.5 mean months vs. 77.0 mean months, *p* < 0.001) [33]. Matsumura et al. discovered another miRNA, miR-19a, as a prognostic biomarker for recurrence. Previously reported to promote proliferation and invasion of cancer cells [34], mir-19a was first identified by the authors to have increased expression in the exosomes of CRC patients with recurrence compared with patients without recurrence and increased expression in CRC patients compared with HCs across stages I–IV. Furthermore, high exosomal miR-19a expression in CRC patients was associated with worse overall survival (OS) and disease-free survival (DSF) than low expression and was an independent risk factor for OS and DFS [35].

Several other miRNAs have also been associated with poor survival and these include miR-21, miR-27a, miR-130a, miR-6803-5p, and miR-221 [31,36,37,38]. In all these studies, exosomal miRNA expression levels were higher in CRC compared with HCs, and when stratified into high and low exosomal expression groups, patients in the high-expression groups had worse OS for each of these individual miRNAs [31,36,37,38].

Although most studies have reported on the significance of overexpressed RNAs, few studies have focused on the significance of underexpressed RNAs. Downregulation of serum exosomal miR-150-5p was observed in CRC patients compared with HCs and exosomal expression of miR-150-5p is significantly higher following surgery compared with pre-resection levels. When survival was assessed, patients with low exosomal expression of miR-150-5p had worse DFS and OS. Furthermore, low serum exosomal miR-150-5p expression, along with lymph node metastasis and TNM stage, were independent prognostic factors for OS [39]. It is interesting to note that a high level of miR 150-5p in the tumor tissue is associated with aggressiveness in triple-negative breast cancer [40]. Gene expression levels in tumor tissue and peripheral blood exosomes may not be directly correlated and understanding these differences is essential to predict the results of exosome gene analysis. Additionally, miR-548c-5p is underexpressed in serum exosomes of CRC patients compared with HCs. On further analysis, patients with liver metastasis compared with no liver metastasis and patients with stage III/IV disease compared with stage I/II disease all had significantly lower miR-548-5p exosomal expression. Compared with patients with high levels of exosomal miR-548c-5p, low exosomal levels was independently associated with worse OS on multivariate analysis [41].

Along with the above-mentioned miRNAs, several lncRNAs have also been identified to have diagnostic and prognostic significance when underexpressed. GAS5 was identified by Liu et al. as a diagnostic and prognostic biomarker along with miR-221; however, unlike miR-221, GAS5 is downregulated in serum exosomes of CRC patients compared with HCs. GAS5 had an AUC of 0.964 for detecting CRC. Low exosomal GAS5 expression was associated with worse OS and was also an independent risk factor for OS [37]. HOTTIP is another lncRNA with significantly decreased expression in serum derived exosomes of CRC patients compared with HCs (AUC = 0.71). Moreover, OS was significantly decreased in patients with low/intermediate expression of HOTTIP (<75th percentile) compared with those with high (≥75th percentile) expression (47.0 months vs. 80.4 months, *p* = 0.0009). On multivariate analysis, low expression of HOTTIP was associated with worse OS [42]. However, it is important to note that there are studies of lncRNAs in which overexpression of lncRNAs was associated with poor outcomes. For example, a study by Liu et al. showed that patients with high expression of exosomal lncRNA CRNDE-h had lower OS rates than the low expression group (34.6% vs. 68.2%, *p* < 0.001) [25]. Collectively, these observations suggest that both over and underexpressed RNAs in the exosomal cargo bear diagnostic and prognostic potential. Identifying the right combinations of markers that provide the most actionable information and insights are essential to develop exosome liquid biopsy as a diagnostic tool for clinical applications. 

**Table 2 cancers-15-01263-t002:** Summary of prognostic exosomal biomarkers for colorectal cancer.

Author(s)	Year	Biomarker(s)	Source	Findings
Matsumura et al. [35]	2015	miR-19a	Serum	Significantly increased levels in CRC patients across stage I-IV disease. High exosomal miR-19a expression was associated with worse OS and DFS
Liu et al. [33]	2016	miR-4772-3p	Serum	Compared recurrent with non-recurrent stage II/III CRC miR-4772-3p was overexpressed in patients with recurrent stage II/III CRC and is an independent predictor of recurrence (AUC = 0.72) and OS
* Liu et al. [25]	2016	lncRNA CRNDE-h	Serum	Significantly higher expression in CRC patients than those with benign colon diseases and HCs AUC = 0.892 to discriminate CRC patients; AUC = 0.913 when combined with CEA High expression associated with lower OS rates
* Tsukamoto et al. [31]	2017	miR-21	Plasma	Higher expression in CRC patients than HCs and increased with increasing cancer stage High exosomal miR-21 expression group had worse OS and DFS than low expression group, specifically among patients with stage II-IV CRC
* Fu et al. [30]	2018	miR-17-5p, miR-92a-3p	Serum	Compared CRC patients with or without metastasis with HCs Significantly higher levels of miR-17-5p and miR-92a-3p in CRC patients with non-metastatic and metastatic disease miR-17-5p AUC = 0.841, miR-92-3p AUC = 0.854 Increased expression correlated with worse pathologic stage
* Yan et al. [36]	2018	miR-6803-5p	Serum	Higher expression in CRC patients Worse OS and DFS in patients with higher levels of exosomal miR-6803-5p High expression was an independent predictor of poor prognosis
* Liu et al. [38]	2018	miR-27a, miR-130a	Plasma	Both exosomal miRNAs are overexpressed in CRC patients AUC = 0.846 for distinguishing stage I CRC patients from HCs High expression was associated with worse OS for both miRNAs
* Liu et al. [37]	2018	lncRNA GAS5, miR-221	Serum	miR-221 was overexpressed in CRC and is an independent predictor of OS GAS5 was underexpressed in CRC and is independent predictor of OS
* Peng et al. [41]	2018	miR-584c-5p	Serum	Underexpressed in CRC patients Lower expression in patients with liver metastases and stage III/IV disease Low expression is an independent factor for worse OS
* Zou et al. [39]	2019	miR-150-5p	Serum	Lower expression levels in CRC patients Expression levels increased after surgical resection CRC patients with low expression had worse DFS and OS Low miR-150-5p expression is an independent prognostic factor of worse OS
* Oehme et al. [42]	2019	lncRNA HOTTIP	Serum	Significantly decreased levels in CRC patients (AUC = 0.71) Low expression of HOTTIP is an independent prognostic marker for worse OS

* These biomarkers were also found to have diagnostic roles. CRC = colorectal cancer, HC = healthy control, OS = overall survival, DFS = disease free survival, CEA = carcinoembryonic antigen.

### 2.3. Predictive Biomarkers

There have been several studies highlighting the potential of RNAs to predict chemoresistance (Table 3). In a study by Jin et al., a panel of four exosomal miRNAs were identified (miR-21-5p, miR-1246, miR-1229-5p, and miR-96-5p) that were upregulated in patients with unresectable stage III/IV CRC who had chemoresistance to 5-FU and oxaliplatin compared with patients who were chemosensitive. This miRNA panel was also able to distinguish chemoresistant CRC patients from their chemosensitive counterparts with an AUC of 0.804 [43].

In another study, Yagi et al. evaluated exosomal miR-125b expression in patients with advanced or recurrent CRC treated with modified fluorouracil, leucovorin, and oxaliplatin (mFOLFOX6)-based first-line therapy before treatment and at different time points during treatment. The response to chemotherapy was evaluated using Response Evaluation Criteria in Solid Tumors (RECIST) and patients were classified into complete response (CR), partial response (PR), stable disease (SD), and progressive disease (PD) groups. The authors found exosomal miR-125b expression post-treatment was significantly lower in patients with PR but significantly higher in patients with PD compared with pre-treatment levels. Patients with SD had no change in exosomal miR-125b expression with treatment. Furthermore, patients in the high miR-125b expression group had worse progression free survival (PFS) than the low expression group. On Cox multivariate regression analysis, KRAS mutation and exosomal miR-125b were found to be independent predictors of PFS. Taken together, these findings suggest miR-125b has the potential to be both a predictive and a prognostic biomarker of CRC [44]. The predictive capacity of exosomes to assess treatment resistance and tumor progression are of remarkable interest in the growing paradigm of utilizing liquid biopsies to dynamically adjust treatment strategies.

**Table 3 cancers-15-01263-t003:** Summary of predictive exosomal biomarkers for colorectal cancer.

Author(s)	Year	Biomarker(s)	Source	Findings
Jin et al. [43]	2019	4 miRNA panel: miR-21-5p, miR-1246, miR-1229-5p, miR-96-5p	Serum	Compared patients with stage III/IV CRC with chemoresistance vs. chemosensitivity Overexpressed in patients with unresectable stage III/IV CRC with chemoresistance compared with chemosensitive counterparts Panel able to distinguish chemoresistant CRC patients from chemosensitive patients with AUC of 0.804
* Yagi et al. [44]	2019	miR-125b	Plasma	Compared CRC patients with HCs and CRC patients with PD vs. PR vs. SD following chemotherapy Expression was significantly higher in patients with CRC than HCs Higher expression with PD and lower in patients with PR following treatment but no change in expression with treatment in patients with SD Patients with high ex-miR-125b expression had worse PFS KRAS mutation and exosomal miR-125b were independent predictors of PFS

* This biomarker was also found to have a prognostic role. CRC = colorectal cancer, HC = healthy control, PD = progressive disease, PR = partial response, SD = stable disease, PFS = progression free survival.

## 3. Gastric Cancer

The number of new GC cases in the US was estimated to be 26,380 in 2022 with 11,090 new deaths [17]. Early-stage GC is often asymptomatic, resulting in a delay in diagnosis. Even with advances in the medical and surgical management of GC over the past decade, late-stage GC is associated with a poor prognosis [12]. This highlights the need for novel noninvasive biomarkers for GC. Here, we discuss the research being conducted on exosomes and the diagnostic and prognostic roles they play in the management of GC (summarized in Table 4 and Table 5).

### 3.1. Diagnostic Biomarkers

As most of the current diagnostic biomarkers for GC are not accurate enough to screen patients for GC, this raises the need to develop new and effective diagnostic biomarkers for the detection of this disease [10]. In 2017, Pan et al. carried out a study analyzing the sera of 94 GC patients and found a significant increase in ZFAS1 expression in GC patient serum exosomes compared with HCs (*p* < 0.001) [45]. Subsequently, another study found a significant increase (2.5×) in the expression of exosomal miR-221 in the peripheral blood of GC patients compared with HCs [52].

Certain miRNAs have been found to have increased diagnostic power when combined. Wang et al., for example, studied the serum of GC patients (*n* = 130) and HCs (*n* = 130) and found a significant increase in the exosomal expression levels of miR-106a-5p and miR-19b-3p in the serum of GC patients (*p* < 0.0001). The diagnostic accuracy was better when the two miRNAs were combined compared with either one of them individually and significantly outperformed the tumor markers alpha fetoprotein (AFP) and cancer antigen (CA) 19-9 [46]. In a study by Huang et al., the serum exosomal expression levels of miR10b-5p, miR132-3p, miR185-5p, miR195-5p, miR20a-3p, and miR296-5p were elevated in GC patients compared with HCs. Integrating the six miRNAs together led to improved accuracy in correctly identifying GC patients with an AUC of 0.703 [54].

Furthermore, as seen in CRC, exosomal biomarkers can be combined with current tumor markers to increase the overall accuracy in screening patients with GC. In the training phase of a study by Ge et al., there was an elevated expression of miR-1307-3p, piR-018569, piR-004918, and piR-019308 in the serum exosomes of GC patients. Evaluation with an additional cohort of patients confirmed significant exosomal overexpression of these miRNAs in GC patient serum (*p* < 0.0001). The diagnostic potential of miR-1307-3p, piR-019308, piR-004918, and piR-018569 was shown with AUCs of 0.845, 0.820, 0.754, and 0.732, respectively. After integration of tumor markers such as CEA and CA 19-9 into the previous biomarkers, their AUCs improved to 0.902, 0.914, 0.859, and 0.868, respectively [59]. Other studies have identified miRNA biomarkers that have lower expression in GC compared with HCs, such as miR-23b [56].

Aside from miRNAs, lncRNAs and circRNAs have also been implicated as diagnostic biomarkers for GC. Cai et al. studied the blood samples of GC and healthy patients and found a significantly lower expression of the lncRNA PCSK2-2:1 in the serum exosomes of GC patients (*p* = 0.006). Serum exosome levels of lncRNA PCSK2-2:1 was additionally found to be accurate in detecting GC patients with an AUC of 0.896 [50]. Zhao et al. studied the serum of 246 GC and healthy patients and found an upregulation in the expression of exosomal lncRNA HOTTIP in GC patients (*p* < 0.001). ROC curve analysis was performed for HOTTIP in patient serum, showing an AUC of 0.827, which was higher than the AUC of other tumor markers (CEA and CA 19-9) combined (*p* < 0.001) [55]. Xie et al. showed the presence of increased expression of circSHKBP1 in the serum of GC patients compared with HCs. These findings were supported with sub-analysis that showed a decrease in the expression of circSHKBP1 after tumor removal in 12 patients in this study [58]. Shao et al. studied the plasma of GC patients, which showed a significant downregulation of Hsa_circ_0065149 in early GC (stage I and II) patient plasma exosomes compared with HCs (*p* < 0.001), with an AUC of 0.640 (*p* = 0.031) by ROC [51]. This, along with previous studies, shows the possibility of increased accuracy of GC detection with the use of exosomal biomarkers compared with other noninvasive biomarkers and tumor markers.

Few studies have specifically evaluated exosomal biomarkers to discriminate GC from other non-malignant conditions of the stomach such as chronic atrophic gastritis (CAG) and intestinal metaplasia (IM). In a study by Lin et al. that evaluated patients with early-stage GC (stage I and II), exosomal lncUEGC1 and lncUEGC2 levels were significantly increased in early-stage GC compared with HCs (*p* < 0.0001). Additionally, plasma exosomal lncUEGC1 expression was significantly increased in stage 1 GC patients compared with CAG patients [49]. A subsequent study that looked at the serum of 862 patients showed significantly elevated circulating levels of exosomal lncRNA-GC1 in GC patients. This biomarker was highly accurate in differentiating GC patients from HCs with an AUC of 0.9033, which was higher than that of current tumor markers such as CEA and CA 19-9. In the verification phase of this study, the circulating exosomal lncRNA-GC1 levels were significantly higher in GC patients compared with those with CAG, IM, and Helicobacter pylori positivity or negativity [57].

### 3.2. Prognostic Biomarkers

Recurrence and metastasis are associated with progression of disease and carry a worse prognosis in GC patients, highlighting the need for prognostic biomarkers that can be used in the monitoring and surveillance of GC patients. Studies that have shown the diagnostic potential of exosomes have given us insights into the prognostic possibilities, as well. In the 2017 study by Pan et al., GC patients were separated into high and low expression of ZFAS1 in patient serum exosomes and the high-expression group was positively associated with lymphatic metastasis (*p* = 0.005) and advanced TNM stage (*p* = 0.010) [45]. Around this time, Ma et al. showed that the serum expression level of miR-221 was associated with TNM stage and overall poor clinical prognosis in GC patients [52]. In the same year, Yen et al. studied the exosome profile in the peripheral blood of 61 GC patients, which showed a positive association between the expression of TGF-β1 and TNM stage (*p* = 0.03) and lymph node metastasis (*p* = 0.01). Additionally, there was a higher level of exosomal TGF-β1 in late-stage GC compared with stage I GC patients and a two-fold increase in patients with lymph node metastasis versus those without [53]. In studies by Wang et al. and Huang et al., the expression levels of different miRNAs (Table 5) were found to be significantly higher in the serum of GC patients with lymphatic metastasis (*p* = 0.001, *p* = 0.008, respectively) and late-stage GC (III and IV) compared with early-stage GC (I and II) (*p* = 0.048, *p* = 0.031, respectively) [46,54].

Studies have also shown the value of exosomal prognostic biomarkers in differentiating between early- and late-stage GC, particularly metastatic disease from non-metastatic disease. Kumata et al. found that exosomal miR-23b levels decreased with the progression of GC with a significant decrease in the expression of miR-23b in stage IV GC compared with earlier stages (*p* < 0.05). After separating patients into high and low miR-23b expression groups, they found an association between miR-23b expression and tumor size, invasion depth, liver metastasis, and TNM stage. Low expression of miR-23b was associated with worse DFS in patients undergoing curative surgery and correlated with recurrence and poor prognosis across all stages of GC [56].

Utilizing a separate set of miRNAs, Zhang et al. found that the expression levels of miR-10b-5p, miR 143-5p, and miR 101-3p were significantly elevated in GC patients with lymph node, liver, and ovarian metastasis, respectively (*p* < 0.05). Additionally, after ROC analysis, the AUCs were calculated to be 0.8919, 0.8247, and 0.8905, respectively (*p* < 0.05) [60].

lncRNAs and circRNAs, have prognostic value as exosomal biomarkers for GC, as well. Although the lncRNA PCSK2-2:1 was significantly lower in the serum exosomes of GC patients, as previously described, it was also correlated with tumor size (*p* = 0.0441), tumor stage (*p* = 0.0061), and the degree of venous invasion (*p* = 0.0367) [50]. Guo et al. were able to show a significant difference in the exosomal levels of lncRNA-GC1 between the four clinical stages of GC with an incremental increase in the exosomal levels of lncRNA-GC1 from stages T1 to T4 and N0 to N3 [57]. Furthermore, Xie et al. showed that the expression of exosomal circSHKBP1 was correlated with advanced TNM stage, vascular invasion, and overall poor prognosis [58]. Studies investigating the field of exosomes have shown promise in the use of exosomes as prognostic biomarkers in the management of GC. Although the wide variation of biomarkers between studies requires further refinement and standardization, the current evidence has shown the potential of exosome cargo to assess disease severity and prognosis in GC.

## 4. Peritoneal Fluid Exosomes in Colon and Gastric Cancer 

Besides blood, peritoneal fluid is of significant interest in GI cancers both due to the propensity of these cancers to metastasize to the peritoneum and the presence of exosomes in the peritoneal space that could be diagnostic of cancer. Moreover, exosomes can play a significant role in creating a premetastatic niche in the peritoneal cavity [61]. The current diagnostic modality to identify microscopic peritoneal disease, namely peritoneal cytology, has a sensitivity as low as 11% to 80% in GC [62]. Hence, exosomal analysis of the peritoneal fluid could be of major importance to fill a critical gap in diagnosing peritoneal disease.

In CRC, Roman-Canal et al. identified exosomal miRNAs from peritoneal lavage of patients with CRC that could potentially have diagnostic significance. Compared with ascites fluid from non-cancer patients, 210 miRNAs were significantly dysregulated in peritoneal lavage fluid from CRC patients. Ten miRNAs were significantly overexpressed in CRC patients with an AUC > 0.95 (Table 6) [22].

In GC, Ohzawa et al. evaluated the miRNA expressions in the peritoneal fluid of patients with and without peritoneal metastasis (PM). Compared with those without PM, the authors found significant upregulation of four miRNAs (miR-21-5p, miR-92a-3p, miR-223-3p, and miR-342-3p) in patients with PM (*p* < 0.05), whereas the miR-29 family, especially miR-29b-3p and miR-29c-3p, was downregulated in all the patients with PM. Additionally, among patients with PM, the expression levels of miR-21-5p, miR-92a-3p, miR-223-3p, and miR-342-3p were positively correlated with the peritoneal carcinomatosis index (PCI) [63]. A follow-up study by Ohzawa et al. confirmed the previous findings, showing decreased expression of the miR-29s (miR-29a-3p, miR-29b-3p, and miR-29c-3p) in the peritoneal lavage fluid/ascites of patients with PM (*p* < 0.001). When patients with T4 tumors status post curative gastrectomy were separated based on low and high expression of miR-29b-3p in the exosomes of the peritoneal fluid, those with low miR-29b-3p expression exhibited worse peritoneal recurrence free survival (RFS) (*p* < 0.05). Furthermore, low expression was associated with significantly worse OS for all three miR-29s (Table 6) [64]. Thus, exosomes isolated from the peritoneal fluid/ascites of GC patients have the potential to be a more accurate alternative to cytology in diagnosing GC patients with PC. These studies highlight the fact that exosomes exist in a variety of biofluids and showcase peritoneal fluid as another possible source of biomarkers for GI cancers.

**Table 6 cancers-15-01263-t006:** Summary of exosomal biomarkers from peritoneal fluid for colorectal and gastric cancer.

Author(s)	Year	Biomarker(s)	Source	Findings
Roman-Canal et al. [22]	2019	miRNA-199b-5p, miRNA-150-5p, miRNA-29c-5p, miRNA-218-5p, miRNA-99a-3p, miRNA-383-5p, miRNA-199a-3p, miRNA-193a-5p, miRNA-10b-5p, miRNA-181c-5p	Peritoneal lavage	Compared peritoneal lavage from CRC patients with ascites from non-cancer patients 210 significantly dysregulated miRNAs were identified from peritoneal lavage of CRC patients Top 10 miRNAs with AUC > 0.95 are listed
Ohzawa et al. [63]	2019	miR-21-5p, miR-92a-3p, miR-233-3p, miR-342-3p	Peritoneal lavage	Compared GC patients with or without PM Increased expression of miR-21-5p, miR-92a-3p, miR-223-3p, and miR-342-3p in peritoneal fluid of patients with PM Expression levels positively correlated with PCI
Ohzawa et al. [64]	2020	miR-29a-3p, miR-29b-3p, miR-29c-3p	Peritoneal lavage	Compared GC patients with or without PM Decreased expression of miR-29s in patients with PM RFS in peritoneum was significantly worse in T4 patients after gastrectomy with low expression of miR-29b-3p Low expression of all three miR-29s was associated with significantly worse OS

CRC = colorectal cancer, GC = gastric cancer, PM = peritoneal metastasis, PCI = peritoneal carcinomatosis index, RFS = recurrence free survival, OS = overall survival.

## 5. Clinical Challenges and Future Prospects

This review highlights the role of exosomes as diagnostic, prognostic, and predictive biomarkers in various GI cancers. However, there are challenges and limitations that need to be overcome before the widespread uptake of exosomal liquid biopsy in clinical practice. First, there are various methods for exosome isolation, each with their own advantages and disadvantages. The two common methods are ultracentrifugation (UC) and precipitation. UC separates exosomes from other components based on size and density differences with high-speed centrifugation [7]. Although UC is considered the gold standard for exosome extraction and separation, UC requires costly instrumentation and is time consuming and more suitable for large volumes, which may not be appropriate in a clinical setting [7]. Precipitation, on the other hand, relies on reducing the solubility of the exosomes, then separating them from other contents using low-speed centrifugation [7]. Compared with UC, the precipitation method requires less time, does not require costly equipment, and can be used with smaller sample volumes. However, other proteins and lipoproteins may be isolated as well, compromising the purity of the exosomes [7]. Current studies often use differing exosome isolation methods, making it difficult to compare studies [8]. Once isolated, EVs should be confirmed as exosomes by examining the morphology, protein expression, size, and concentration using methods proposed by the International Society of Extracellular Vesicles (ISEV) [65], which takes additional time and expertise. Given these challenges, exosome isolation and characterization methods need improved efficiency and throughput to be successfully transitioned to clinical application.

Second, the study of exosomal gene expression requires standardization of its methods and reference genes. In our review, for example, we present many different miRNAs, lncRNAs, circRNAs, and proteins that have been identified as having diagnostic, prognostic, and predictive potential with limited overlap for the same cancer types. There are also contrasting findings between cancer types. For example, low expression of the exosomal lncRNA HOTTIP, as described above, was found to be an independent predictor of worse OS in CRC [42]. These findings contrast with that of a recent meta-analysis by Fan et al. in which HOTTIP tumor tissue overexpression in multiple cancers was associated with increased tumor stage, lymph node metastasis, distant metastasis, and poor OS [66]. Although, the differences between the tumor expression levels of genes and exosomal levels could be different and may account for these observed differences, an exosomal study in GC showed high levels of expression of HOTTIP to be diagnostic of GC [55]. These contradicting results are a prime example of issues in exosome research and are one of the major barriers to clinical translation. Furthermore, standardization of the control group is needed to compare differences in gene expression among the experimental groups. HCs are often used in the studies we describe but it is important to note variability in gene expression can also exist among healthy individuals. Therefore, an accepted set of reference genes with expected expression levels should be developed.

Future work can be conducted to identify universal exosomal genes with known functions in cancer or gene panels that may have more diagnostic, prognostic, or predictive power than individual biomarkers. Another area of exciting application is the use of exosomes in therapeutics as drug delivery systems [67,68]. Therapeutic applications of exosomes are beyond the scope of this review; however, the readers are directed to cited references for detailed information. Exosomal genes with known functions in cancers can also be targeted for therapy in the future.

## 6. Conclusions

In short, exosomes are an exciting and promising source of biomarkers with the potential for clinical application as liquid biopsies for various GI cancers. Exosomes carry a variety of RNAs and proteins, many of which have shown diagnostic, prognostic, and predictive potential. More work is needed to overcome the challenges of applying exosomes to clinical practice. However, the abundance of information that can be garnered from exosomes is apparent and exciting.

## Figures and Tables

**Figure 1 cancers-15-01263-f001:**
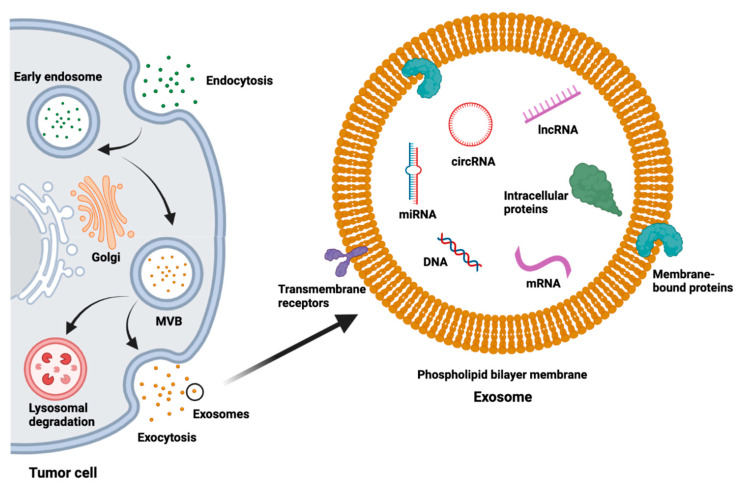
Exosome biogenesis, structure, and contents. MVB = multivesicular body.

**Figure 2 cancers-15-01263-f002:**
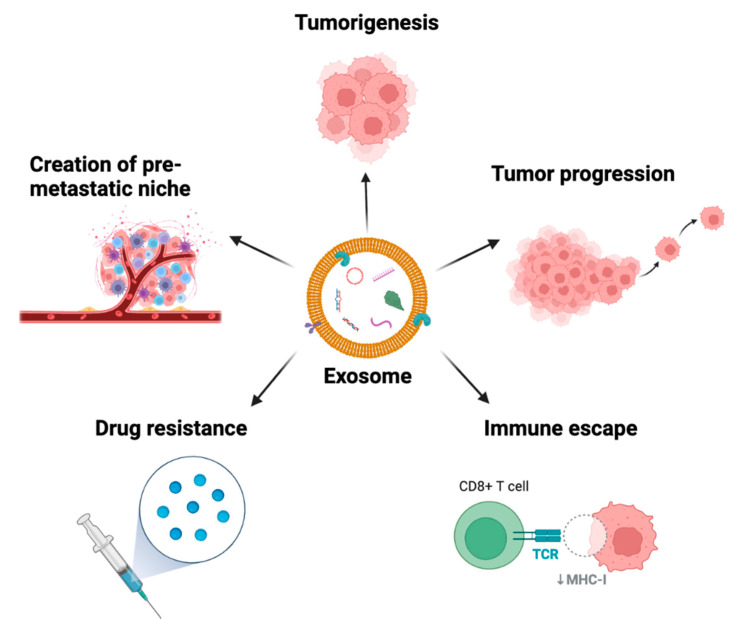
The roles of exosomes in cancer. TCR = T cell receptor, MHC-I = major histocompatibility complex class I.

**Figure 3 cancers-15-01263-f003:**
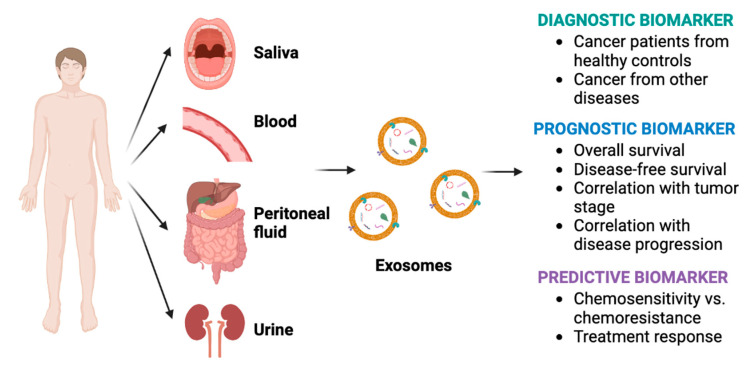
Diagnostic, prognostic, and predictive roles of exosome biomarkers in cancer.

**Table 4 cancers-15-01263-t004:** Summary of diagnostic exosomal biomarkers for gastric cancer.

Author(s)	Year	Biomarker(s)	Source	Findings
* Pan et al. [45]	2017	lncRNA ZFAS1	Serum	ZFAS1 expression is elevated in serum exosomes of GC patients (AUC = 0.837) Increased exosomal expression of ZFAS1 is positively correlated with lymphatic metastasis and worse TNM staging
* Wang et al. [46]	2017	miR-19b-3p, miR-106a-5p	Serum	miR-19b and miR-106a is overexpressed in GC patients AUC for miR-106a-5p is 0.786, AUC for miR-19b-3p is 0.769, combined AUC is 0.814 Expression of miR-106a-5p and miR-19b-3p was correlated to GC lymphatic metastasis and increased in advanced GC stages (III and IV) compared with earlier stages (I and II)
Li et al. [47]	2018	miR-217	Plasma	Increased expression of miR-217 in plasma exosomes of GC patients
Fu et al. [48]	2018	TRIM3, miR-20a	Serum	Decreased expression of TRIM3 in serum exosomes of GC patients TRIM3 is negatively regulated by miR-20a
Lin et al. [49]	2018	lncUEGC1, lncUEGC2	Plasma	Compared early GC patients (stage I and II) with CAG patients to HCs Overexpression of lncUEGC1 in early GC patients lncUEGC1 discriminated early GC from HCs (AUC = 0.8760) and CAG (AUC = 0.8406)
* Cai et al. [50]	2019	lnc RNA PCSK2-2:1	Serum	Downregulation of lncRNA PCSK2-2:1 expression level in serum exosomes of GC patients AUC for lncRNA PCSK2-2:1 was 0.896 Expression of lncRNA PCSK2-2:1 was correlated with tumor size, tumor stage, and venous invasion
Shao et al. [51]	2020	Hsa_circ_0065149	Plasma	Compared early GC patients (stage I and II) to HCs Decreased expression of Hsa_circ_0065149 in plasma exosomes of early GC patients (stage I and II)

* These biomarkers were also found to have prognostic roles. GC = gastric cancer, HC = healthy control, CAG = chronic atrophic gastritis.

**Table 5 cancers-15-01263-t005:** Summary of prognostic exosomal biomarkers for gastric cancer.

Author(s)	Year	Biomarker(s)	Source	Findings
* Ma et al. [52]	2017	miR-221	Serum	Increased expression of miR-221 in the peripheral blood of GC patients Increase expression of miR-221 is positively correlated with poor clinical prognosis (TNM stage)
Yen et al. [53]	2017	TGF-β1	Serum	Exosomal TGF-β1 expression is related to lymph node metastasis and advanced TNM staging
* Huang et al. [54]	2017	miR10b-5p, miR132-3p, miR185-5p, miR195-5p, miR20a-3p, miR296-5p	Serum	All 6 miRNAs were overexpressed in serum exosomes of GC patients with miR10b-5p, miR195-5p, miR20a-3p, miR296-5p having significant overexpression Combined AUC for 6 miRNAs is 0.703 in serum Increased serum expression of miR10b-5p, miR185-5p, and miR296-5p in late-stage GC (III and IV) compared with early-stage GC (I and II)
* Zhao et al. [55]	2018	lncRNA HOTTIP	Serum	Increased expression levels of exosomal HOTTIP in GC patients AUC for HOTTIP is 0.827 Expression levels of exosomal HOTTIP was correlated with invasion depth, TNM stage and poor OS
* Kumata et al. [56]	2018	miRNA-23b	Plasma	Decreased expression of exosomal miR-23b levels in GC patients Expression level of miR-23b was correlated with tumor size, depth of invasion, liver metastasis and TNM staging Low expression of miR-23b was associated with recurrence and worse OS compared with high expression of miR-23b
* Guo et al. [57]	2020	lncRNA-GC1	Serum	Increased exosomal expression of lncRNA-GC1 in GC patients (AUC = 0.9033) Levels of circulating exosomal lncRNA-GC1 were significantly associated with GC from early to advanced stage
* Xie et al. [58]	2020	circSHKBP1	Serum	Elevated expression of circSHKBP1 in serum exosomes of GC patients Increased expression of circSHKBP1 was related to advanced TNM stage, vascular invasion, and poor survival
* Ge et al. [59]	2020	miR-1307-3p, piR-019308, piR-004918, piR-018569	Serum	Compared GC patients with HCs to patients with CAG, IM, or H. pylori Increased expression of miR-1307-3p, piR-019308, piR-004918, piR-018569 in serum exosomes of GC patients AUC for miR-1307-3p, piR-019308, piR-004918, piR-018569 is 0.845, 0.820, 0.754, and 0.732 Higher expression of piR-004918 and piR-019308 in GC patients with metastatic disease
Zhang et al. [60]	2020	miR-10b-5p, miR-101-3p, miR-143-5p	Plasma	Compared GC patients with GC patients with lymph node, ovarian, or liver metastasis Expression levels of miR-10b-5p, miR-101-3p, and miR-143-5p were related to lymph node, ovarian, and liver metastasis AUC for metastasis > 0.82 for all three miRNAs

* These biomarkers were also found to have diagnostic roles. GC = gastric cancer, HC = healthy control, OS = overall survival, CAG = chronic atrophic gastritis, IM = intestinal metaplasia, H. pylori = Helicobacter pylori.

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
