# Peer review of "Exosomes as a Source of Biomarkers for Gastrointestinal Cancers"

_cancers, 2023, doi:10.3390/cancers15041263_

Round 1

Reviewer 1 Report

Exosomes as Source of Biomarkers for Gastrointestinal Cancers, a work by Yu et al. declared that liquid biopsies have become an integral part of cancer management due to their application in early diagnosis, molecular profiling, disease surveillance and treatment response monitoring. Exosomes have emerged as a promising platform for liquid biopsy given their abundance in various body fluids including blood, saliva, and urine and their role in cancer development and progression. Exosomes are small, extracellular vesicles, nanometers in size that are rich in tumor-derived material such as DNA, RNA, and proteins. In the past decade, numerous studies have been published about the functional significance of exosomes in a wide variety of cancers, with a particular focus on exosome-derived RNAs and proteins. In this review, utilizing human studies on exosomes, we highlight their potential as diagnostic, prognostic, and predictive biomarkers in gastrointestinal cancers. The finding is interesting, however the abstract section should be improved. I provided some comments on the manuscript. Concerns should be addressed by authors.

1.     English grammar and typo errors must to be corrected.

2.     Rephrase and better the abstract.

3.     Exosomes are 50-150 nm EVs.

4.     Indicate EVs at the first time of mention.

5.     Cite relevant references (  PMID: 36604402; PMID: 35427569)

6.     Provide a figure about exosomes release from tumor cells.

Reviewer 2 Report

The focus of this work is interesting. Considering the lack of information regarding the role of exosome, your review provides an excellent view upon this field.

Not only the analysis of up- and downregulation of the exosome through the literature is well conducted and really attractive, but also the combination with the study of “common” biomarkers represents an interesting point of view.

It is obvious the importance of the role of exosomal expression into the diagnosis of gastric cancer, being this one often misdiagnosed. The delay of the diagnosis provides worst overall survival and even higher cost for the sanitary system.

The information that these biomarkers could give are extremely interesting. The details regarding the depth of tissue and vascular invasion and the presence of metastasis could be predictable especially in those cases where the staging is unclear.

Regarding this, although the sensitivity of the study of peritoneal cytology is extremely variable (from 11% to 80% in gastric cancer), the individuation of exosome in peritoneal cavity in those patients with a CUP could provide additional information.

The recent literature, more and more often, has been focusing on the exosome as therapeutic tool. Engineering could help to utilize exosomes as target of medical therapy. Huarui Zhang et al, but also Zijie Xu more recently, have highlighted the development of exosome as a drug delivery system. What is your opinion about this therapeutic chance?

Also, there are studies about the expression of exosomes in patients that had synchronous or metachronous tumors?
